# Caregiver Feeding Practices as Predictors for Child Dietary Intake in Low-Income, Appalachian Communities

**DOI:** 10.3390/nu13082773

**Published:** 2021-08-13

**Authors:** Mikaela B. McIver, Sarah Colby, Melissa Hansen-Petrik, Elizabeth T. Anderson Steeves

**Affiliations:** Department of Nutrition, University of Tennessee, Knoxville, TN 37996-1920, USA; mikaela.mciver@knoxcounty.org (M.B.M.); scolby1@utk.edu (S.C.); phansen@utk.edu (M.H.-P.)

**Keywords:** feeding practices, childhood, health disparities, Appalachian, rural, dietary intake, obesity, low-income

## Abstract

The Appalachian region of the U.S. is disproportionately impacted by poverty, obesity, and nutrition-related chronic diseases. Evidence suggests that caregiver feeding practices may promote healthful eating behaviors among children; however, this has not been examined in low-income, rural, Appalachian populations. This study examines caregiver feeding practices as predictors for child diet in low-income Appalachian families, using a cross-sectional analysis of 178 caregivers of young children (ages 2–10 years old), that were recruited from low-income, rural communities in East Tennessee, from November 2017 to June 2018. Caregivers self-reported measures of demographics, feeding practices, and child dietary intake. Multiple linear regression analyses were run, and found that higher use of caregiver modeling positively predicted child vegetable consumption (Beta = 1.02; *p* = 0.04). Higher caregiver intake of fruits and vegetables positively predicted child fruit consumption (Beta = 0.29; *p* = 0.02) and vegetable consumption (Beta = 1.56; *p* < 0.001), respectively. Higher home availability of healthier foods positively predicted child fruit consumption (Beta = 0.06; *p* = 0.002) and vegetable consumption (Beta = 0.09; *p* = 0.05). Higher home availability of less healthy foods positively predicted child consumption of high-sugar/high-fat snack foods (Beta = 0.59; *p* = 0.003). The findings of this study indicate that caregiver modeling, healthy caregiver dietary intake, and healthful home food availability are associated with healthier child dietary intake among young children in low-income, rural, Appalachian areas. Promoting these practices among caregivers may be an important strategy to enhancing dietary intake of children in this hard-to-reach, underserved population.

## 1. Introduction

Dietary recommendations for children in the United States are based upon the Dietary Guidelines for Americans, which provide evidence-based recommendations for foods and beverages to consume to promote nutritional health and reduce risk of chronic disease [1]. However, few children in the United States meet these federal dietary recommendations [2]. Sixty percent and 93% of children fall short of recommendations for fruits and vegetables, respectively [3]. The majority of children also consume excess energy from solid fat and added sugars [2]. Unhealthy eating patterns are associated with excessive weight gain and may be predictive of disease risk and overall health status [4]. To prevent excessive weight-gain and reduce the risk of chronic disease, it is recommended that children consume a diet rich in fruits and vegetables and limit consumption of added sugars and saturated fat [4].

Parenting styles are associated with child dietary intake, as evidenced both by cross sectional and longitudinal studies [5,6,7]. Parenting styles that are commonly referenced in the literature include authoritative, authoritarian, permissive, and neglectful styles [5,6]. Specifically, the authoritative parenting style, which is characterized by high levels of caregiver warmth and control over the feeding situation, is associated with healthier dietary intake among children [5,6].

Caregiver feeding practices and home environmental factors including caregiver role modeling of dietary behaviors, caregiver dietary intake, and home food availability have been found to be predictive of child diet in certain populations, such as those of higher-income, higher educational attainment, and urban populations [8]. One systematic review of 37 studies of caregiver feeding practices found that caregiver role modeling, home food availability, and caregiver dietary intake were important predictors of child dietary intake for both healthy and unhealthy foods. However, the effectiveness of feeding practices may vary according to child age and parenting style. For example, caregiver feeding practices that are known to promote healthy dietary intake among children, when used in the context of different parenting styles may have different effects [8]. Additionally, rewarding with verbal praise predicted child diet most strongly among children aged 6 and younger, while modeling and home food availability appear to predict food consumption among children aged 2–11 years old [8], thus examining caregiver feeding practices, such as parental modeling and home food availability in diverse population and settings is critical. In addition, the literature suggests that educating caregivers of young children about the use of caregiver feeding practices may promote healthy eating and prevent unhealthy eating [9], thus having evidence that demonstrates this along underserved populations, such as low-income, rural, Appalachian families may be a helpful strategy to reduce the nutrition-related health disparities seen in this population.

Caregiver role modeling of dietary behaviors (from here on referred to as modeling) is rooted in Bandura’s social cognitive theory and posits that children’s observations of caregiver eating behaviors can influence child diet [10,11]. Previous studies among children and adolescents aged 2–18 have defined modeling in two ways: (1) the level of importance caregivers place on modeling healthy eating behaviors and the frequency with which caregivers report these behaviors [9,12,13,14,15,16], and (2) caregiver dietary intake of specific foods [17,18]. The first definition of modeling captures caregiver’s reported dietary intake behaviors in addition to social factors, caregivers’ food-related attitudes, and behaviors around meal times, and is, therefore, used most commonly in the literature. Using this definition, modeling was positively associated with child dietary intake of fruits and vegetables, lower consumption of sugar sweetened beverages, sweets, and snacks, and is inversely associated with a child’s BMI z-score [9,12,13,14,15,16,18,19,20,21]. Fewer studies have assessed the modeling construct as caregiver dietary intake of specific foods [17,22]. In one study, healthier caregiver dietary intake was found to be positively associated with adolescent consumption of fruits and vegetables and negatively associated with sugar sweetened beverage consumption [18]. Knowing the impact of both of these concepts of caregiver modeling among low-income, rural, Appalachian youths could be a key strategy to improving nutritional health in this population.

Home food availability refers to caregiver control over the types of food available at home. Previous studies have linked the availability of healthier food at home to higher child consumption of fruits and vegetables [13,17,23,24] and lower consumption of high-sugar/high-fat (HS/HF) snack foods among children in higher income, more educated, and urban populations [17,23], indicating that the availability of healthier foods in the home may play a role in developing child preferences for healthier foods in certain groups [8]. Further, low fruit and vegetable consumption among children has been found to be associated with low availability of fruits and vegetables in the home and low caregiver socio-economic status [25].

The Appalachian region is geographically located in the Eastern United States, surrounding the Appalachian Mountains. This region has a higher than average rural population and adult obesity and chronic disease rates, such as diabetes and cardiovascular disease, that exceed national averages [26,27]. Historically, the Appalachian region has been encumbered by high rates of poverty [28]. Despite recent progress, the region as a whole continues to experience higher than national averages for both poverty and unemployment rates, exacerbating health disparities between Appalachian communities and other regions of the United States [29].

The prevalence of obesity is higher among both low-income [30] and rural populations [31], and rural youths have 26–30% higher odds of obesity than urban youths, even after controlling for sociodemographic factors, health, diet, and exercise behaviors [31,32]. According to a study of demographic characteristics and diet quality, individuals with low socio-economic status are less likely to adhere to federal dietary recommendations [33]. Furthermore, children living in rural areas tend to have poor adherence to dietary patterns compared to non-rural children [34]. For example, according to national data, rural children consume an average of 90 more kilocalories per day and are less likely to consume any fruit or meet the daily recommendation for fruit when compared to urban children [32].

Rural, Appalachian communities are at a high risk for poor diet quality and diet-related health disparities, and, therefore, should be considered as an important sub-population in future research. Specifically, little is known regarding this population’s use of feeding practices and how each of these factors relates to a child’s consumption of fruit, vegetables, and high-sugar/high-fat snack foods (e.g., candy, doughnuts, cookies, ice cream).

Previous studies of caregiver feeding practices have been conducted in non-rural settings and among higher income populations [9,13,14,15,16,18], thus limiting the generalizability of findings to this population. The aims of this study, therefore, were to describe the use of modeling, caregiver dietary intake, and home food availability; and to examine associations between modeling, caregiver dietary intake, and home food availability with child fruit consumption, child vegetable consumption, and child HS/HF snack food consumption, among families with young children in low-income, rural areas in Appalachian East Tennessee.

## 2. Materials and Methods

### 2.1. Study Design and Participant Eligibility Criteria

This study was an extension of another study, Shop Smart Tennessee (SSTN) [35]. SSTN was a multi-level intervention implemented in six low-income, rural, Appalachian communities that aimed to increase both access to and demand for healthier items in convenience stores through store-based intervention strategies such as storeowner training, increased stocking of healthier foods, in-store marketing (signage and shelf-labels), and promotions (taste tests, recipes, giveaways). This study analyses cross-sectional data from the baseline sample of SSTN that was collected from November 2017 to June 2018. Hypotheses and the data analysis plan for this study were specified prior to data collection and are independent from the aims of the SSTN study. Data related to home food availability, modeling, and caregiver and child dietary intake were collected from caregivers via an interviewer-administered survey. Inclusion criteria for SSTN and the present study required that participants were the primary caregiver of a child aged 2–10 years old (with caregiver being inclusively defined as an adult who provided care for the child at least 50% of the time, and could include parents, step-parents, grandparents, foster parents, or others who self-identify in this role), were the primary food shopper for their household, were over the age of 18 themselves, and regularly shopped at a food retail store participating in the parent study (≥one time per week). One caregiver/child dyad per household was eligible to participate. When caregivers had more than one eligible child in their household, they were asked to select the child with dietary habits that they felt they were most familiar with. For example, caregivers may be more familiar with the dietary intake of a younger child versus an older child who receives most of their meals at school. Potential participants were excluded if they did not meet all of the above eligibility criteria.

### 2.2. Recruitment and Data Collection

Caregivers were recruited from food retail stores across six low-income communities in rural, Appalachian East Tennessee through in-person recruitment. The research team visited participating stores weekly to recruit caregivers, screen potential participants for eligibility, and collect contact information from individuals who expressed interest in participating. The research team also placed recruitment materials (flyers, posters, table tents, and other signage) in stores and at nearby community locations. After caregivers enrolled in the study, surveys were administered either in-person at the point of recruitment or by phone according to participant availability. Regardless of whether the survey was administered in-person or over the phone, data collectors administered the survey verbally, reading the questions aloud to the participant and recording their responses. Data were collected by trained, graduate-level nutrition students and research staff. Survey administration time was approximately 45 min for both in-person and phone data collection. This study was approved by the University of Tennessee Institutional Review Board (IRB-17-03870-XP). Upon survey completion, each participant received a USD 25 gift card.

### 2.3. Measures

Surveys consisted of caregiver and child sociodemographic questions and a two-item food security screener [36], the HomeSTEAD caregiver modeling questionnaire [16], a modified Home Food Inventory [37] to assess home food availability, the National Cancer Institute’s Dietary Screener Questionnaire (DSQ) [38] to assess child dietary intake, and the Behavioral Risk Factor Surveillance System (BRFSS) fruit and vegetable module [39] to assess caregiver dietary intake. All survey instruments were drawn from existing literature and were pilot tested among a sample of rural caregivers for use in the present study.

Caregiver role modeling was measured with a scale from the HomeSTEAD Family Food Practices survey which has been previously validated for use among caregivers of children 3–12 years old in a Southeastern U.S. population [16]. The modeling scale measures caregiver self-reported role modeling of healthy eating behaviors on a five-point Likert scale (“strongly disagree” (1) to “strongly agree” (5) or “never” (1) to “always” (5)). The scale assessed level of agreement (“I try to eat healthy foods in front of my child, even if they are not my favorite”) and the self-reported frequency of modeling behaviors (“I eat food I want my child to eat”). One item on the scale refers to the frequency of the consumption of less healthy beverages, and this item is reverse scored. See Table 2 for a list of all questions included in this scale and how they are scored. One question was removed from the original six-item scale to increase internal consistency (Cronbach’s *α* = 0.63). Responses were averaged to obtain a mean score for caregiver modeling with a possible range of one to five. A higher score reflects higher use of caregiver modeling [16].

Home food availability was measured using a 59-item modified version of the Home Food Inventory (HFI) [37]. For the purpose of this study, the HFI was tailored to include healthier foods and their less healthy counterpart (e.g., low sugar cereals vs. sugary cereals) identified through formative research among rural, Appalachian caregivers. Items were listed by category with a “yes/no” response option (yes = 1, no = 0). Affirmative answers indicated that the item was present in the household at the time of data collection. Sums were calculated for the availability of healthier items (42 items; Cronbach’s *α* = 0.85) and less healthy items (14 items; Cronbach’s *α* = 0.58). For both sums, a higher score indicates higher availability in the home [37]. A higher score for healthier home food availability and a lower score for less healthy food availability indicate a healthier home food environment.

Child consumption of fruits, vegetables, and HS/HF snack foods was measured using the National Cancer Institute’s Dietary Screener Questionnaire (DSQ) [38]. This 19-item food frequency screener questionnaire measures the frequency of consumption of foods and beverages over the previous 30-day period. On the DSQ, frequency of consumption is reported using categorical responses. Categorical responses for all items assessed in this study were reported on a scale from “never” (0) to “two or more times per day” (8) on average over the past 30-day period [38]. Frequency of consumption for fruit (fresh, frozen, or canned fruit (but not fruit juice) was measured with one item on the DSQ, vegetable intake (lettuce/green salads; non-fried potatoes; and “other” vegetables) was measured with three items from the DSQ, and HS/HF snack food (candy; doughnuts and pastries; cookies, cake, pie, and brownies; and ice cream or other frozen desserts) was measured with four items from the DSQ. Scored responses for each item within each scale (fruit, vegetable, and HS/HF snack food) were summed and analyzed. After sums were created, possible scores for frequency of consumption for fruit, vegetables, and HS/HF snack foods ranged from 0–8, 0–24, and 0–32, respectively, with higher scores indicating more frequent consumption. This scoring method is consistent with how the questions are structured on the DSQ and allowed child fruit, vegetable, and HS/HF snack food consumption to be analyzed separately, and measured in terms of frequency of consumption over the past 30 days.

Caregiver dietary intake was measured using the BRFSS fruit and vegetable module [39]. The module measures the frequency of consumption of all fruit (fresh, frozen, or canned) and vegetables (lettuce/greens salads, non-fried potatoes, and “other” vegetables). Responses were recorded as the number of times per day, week, or month consumed in the previous 30-day period. From these measures, servings per day were calculated.

### 2.4. Statistical Analysis

Following data collection, data were downloaded from the Qualtrics (Qualtrics, Provo, UT, USA) data entry platform, cleaned, and checked for missing or incomplete data or entry errors. Participants with missing data were re-contacted to collect the data, and if they could not be re-contacted, were removed from linear regression models (*n* = 4). Descriptive statistics were run to assess the study population and outcomes of interest.

Three multiple linear regression analyses were conducted to examine the relationships between caregiver feeding practices of interest and child dietary variables (fruit, vegetable, and HS/HF snack food consumption). The first multiple linear regression was calculated to predict child fruit consumption based on modeling, healthier home food availability, and caregiver fruit intake. The second multiple linear regression was calculated to predict child consumption of vegetables based on modeling, healthier home food availability, and caregiver vegetable intake. The third multiple linear regression was calculated to predict child consumption of HS/HF snack foods based on modeling and home food availability of less healthy foods. Caregiver intake of HS/HF snack foods was not measured and, therefore, was not included in statistical models. In each model child age, gender and household income were controlled for, as these factors may influence child dietary intake, as seen in similar studies [14,15,17]. Covariates were entered into the models in two blocks, where independent variables (modeling, caregiver intake [when included in the model], and home food availability) were entered in the first block, then potential confounders (child age, child gender, and household income) were entered in the second block. For each of the three models, assumptions of multiple linear regression were assessed (e.g., tests for normality, multicollinearity, and homoscedasticity were conducted) and all assumptions were met for each of the three models. Data analyses were conducted using SPSS software, version 25 (IBM Corp. Released 2017. IBM SPSS Statistics for Mac, Version 25.0. IBM Corp., Armonk, NY, USA).

## 3. Results

### 3.1. Sample Characteristics

A total of 354 participants screened as eligible for the study, 178 caregiver-child dyads completed baseline assessments, and 174 were included in the final analytic sample (Figure 1). Caregiver participants were primarily female (78%) with an average age of 35.6 (±9.8) years. Close to half of child participants were female (54%) with an average age of 6.5 (±2.7) years. The majority of the sample identified as white (97% caregivers and 96% children) and non-Hispanic (99% caregivers and 98% children). The majority of the sample (66%) reported an annual household income of USD 30,000 or less, which is similar to the Federal Poverty Level for a family of five. The average household size for the sample was 4.4 (±1.8) individuals. Of the sample, 59% of households were food insecure according to a 2-item food insecurity screener [36]. Further descriptive characteristics of the sample are shown in Table 1.

### 3.2. Modeling, Caregiver Dietary Intake, and Home Food Availability

Scores for caregiver use of modeling behaviors and home food availability of healthier and less healthy foods are shown in Table 2. The average score for caregiver modeling was 3.6 (±0.63) on a scale from one to five. A score of 3.6 (±0.63) on the modeling scale indicates that, on average, caregivers reported that they “agree” with statements about their use of modeling behaviors when given a scale of strongly disagree to strongly agree or that they “often” engage in modeling behaviors when given a scale of never to almost always. The average score for home food availability of healthier foods was 20.3 (±7.24) items out of a total possible score of 42.0 healthier items. The average score for less healthy home food availability was 10.1 (±2.35) items out of a total possible score of 14.0 less healthy items.

### 3.3. Caregiver Self-Reported Child and Caregiver Dietary Intake

Caregiver-reported measures of child and caregiver dietary intake are shown in Table 3. The mean child fruit consumption frequency score was 7.1 (±1.78) on a scale from zero to eight, indicating that on average in a 30-day period, children ate fruit about one time per day. The mean child vegetable consumption frequency score was 14.8 (±4.70) on a scale of 0–24, indicating that on average in a 30-day period, children ate vegetables 3–4 times per week. The mean child HS/HF snack food consumption frequency score was 17.6 (±6.33) on a scale of 0–32, indicating that on average in a 30-day period, children ate HS/HF snack foods about 2 times per week. The mean caregiver fruit consumption frequency was 1.0 (±1.09) times per day in a 30-day time period. The mean caregiver vegetable consumption frequency was 1.8 (±1.16) times per day in a 30-day time period.

### 3.4. Multiple Linear Regression Predicting Child Fruit Consumption

After adjusting for confounders (child age, child gender, household income), higher healthier home food availability and caregiver fruit consumption frequency were significant predictors of child fruit consumption frequency (Table 4). Higher availability of healthier foods at home was positively associated with higher child fruit consumption frequency (Beta = 0.06; *p* = 0.002). Similarly, caregivers who reported consuming more fruit had children with significantly higher fruit consumption frequency (Beta = 0.29; *p* = 0.02). Higher modeling was not a significant predictor of higher child fruit consumption. Overall, the full model explained 18% of the variability in child fruit consumption frequency.

### 3.5. Multiple Linear Regression Predicting Child Vegetable Consumption

After adjusting for confounders, modeling, healthier home food availability, and caregiver vegetable consumption frequency were found to be statistically significant predictors of child vegetable consumption frequency (Table 5). Caregivers who reported higher frequency of vegetable consumption had children with significantly higher vegetable consumption frequency (Beta = 1.56; *p* < 0.001). Similarly, modeling was positively associated with higher child vegetable consumption frequency (Beta = 1.02; *p* = 0.04). Healthier home food availability was also a significant predictor of child vegetable consumption (Beta = 0.09; *p* = 0.05). The full model explained 27% of the variability in child vegetable consumption frequency.

### 3.6. Multiple Linear Regression Predicting Child HS/HF Snack Food Consumption

After adjusting for confounders, parental modeling and home food availability of less healthy foods were statistically significant predictors for child HS/HF snack food consumption frequency (Table 6). A greater presence of less healthy foods in the home was significantly associated with higher child HS/HF snack food consumption frequency (Beta = 0.59; *p* = 0.003). Higher modeling scores were inversely related to HS/HF snack food consumption, meaning that modeling of healthier eating behaviors is associated with decreased HS/HF snack food intake, (Beta = −1.43; *p* = 0.05). The full model explained 10% of the variability in child HS/HF snack food consumption frequency.

## 4. Discussion

This study offers a significant contribution to the literature as it is among the first to assess the use of caregiver modeling, caregiver dietary intake, and home food availability as measures of caregiver feeding practices in a rural, Appalachian population sampled from low-income communities. Prior to the completion of this study, little was known about the relationship between these factors and child food consumption in this population. These findings help to identify potential child health promotion strategies for use among low-income, rural Appalachian families.

The mean score of caregiver reports of modeling behaviors in the present study were consistent with cross-sectional findings from a study by Vaughn and colleagues in a non-rural sample of highly educated families with higher-incomes [16]. Despite population differences, the reported use of modeling in this population was found to be similar to previous studies. However, one cross-sectional study found that low-income, rural mothers had poor alignment between their intent to promote healthier child dietary intake and the use of counterproductive feeding practices [40]. Further research is needed to better understand how a low-income, rural context may shape the application of these caregiver feeding practices in interventions aiming to improve child dietary intake.

Caregiver modeling significantly predicted child consumption of vegetables, which is consistent with the current literature for other population groups in cross-sectional studies [9,12,13,14,15,16,17,18]. Modeling also inversely predicted HS/HF snack food consumption, meaning higher levels of parental modeling was associated with lower frequency of HS/HF snack food consumption. There was not a significant relationship identified between modeling and fruit intake in this study. Previous cross-sectional studies have reported that caregiver modeling is a predictor of higher child consumption for both fruits and vegetables [9], and lower consumption of less healthy foods, such as soda or HS/HF snack foods [16,17]. However, studies often assess fruit and vegetable consumption as a combined category, limiting the ability to interpret results. The present study analyzed fruits and vegetables as individual variables, as children’s consumption patterns of fruit and vegetables differ, with the general consensus being that among children, fruit consumption is easier to modify than vegetable consumption [3,41], which is potentially due to a variety of factors such as preferences for the taste and texture of fruit, or that fruit is ready-to-eat and often consumed as a snack [41,42].

Caregiver dietary intake of fruit and vegetables were found to be significant predictors of child fruit and vegetable consumption, respectively. This is also consistent with current literature on caregiver modeling when measured as caregiver dietary intake in cross-sectional studies [17,18]. However, the literature assessing the relationship between caregiver dietary intake and child dietary intake patterns is limited when compared to other food-related parenting factors. While this is an important and novel finding of our study, further research is needed to confirm these findings, and to determine if this finding is generalizable to other groups.

The present study found that higher home availability of healthier foods was positively associated with child fruit consumption, which is consistent with reports from multiple previous cross-sectional studies [13,17,23,24,43,44]. Higher availability of healthier food was positively associated with child vegetable consumption. In a 2014 cross-sectional study by Loth and colleagues, home availability of healthier foods was associated with observed differences in child consumption of both fruits and vegetables [17]. Similarly, another cross-sectional study found that overall higher diet quality, including high intake for both fruits and vegetables, was associated with home availability of healthier foods [14].

Similar to a cross-sectional study by Hendy and colleagues [44], the present study found that home availability of less healthy foods was associated with high child consumption of HS/HF snack foods. Based on this knowledge, limiting the availability of less healthy snack foods in the home may be a useful strategy to limit children’s consumption of HS/HF snack foods. Because children’s preferences develop over time and through multiple exposures to foods [45], promoting a healthier home food environment may influence child dietary intake patterns both inside and outside of the home. While caregiver feeding practices such as home food availability and caregiver modeling may influence child dietary intake both inside and outside of the home, it is important to note that the relationships influencing child obesity and child diet quality are complex and multi-faceted [46]. Therefore, while the percent of variability explained by the regression models in this study ranged from 10% to 27%, which is consistent with the literature [8,41], there is still a lot of variability in child fruit, vegetable, and HF/HS intake that was not explained by these models. Because of this, these results should be considered within the greater context of childhood nutrition interventions and the various factors at play, and additional research is needed to further explore influences on child dietary intakes.

This study is among the first to investigate modeling, caregiver dietary intake, and home food availability in a rural, Appalachian population in low-income communities. Despite the fact that this target population experiences nutrition-related health disparities [26,27], this population is one that can be difficult to reach and may not be well represented in the current literature. The assessment of modeling as two distinct constructs is a novel approach and should be further explored in future research. Finally, a key strength of this study was the separate dietary analysis of fruits and vegetables, which has not been done in much of the previous literature.

Key limitations to this study include the use of a convenience sample and cross-sectional data. The target population of this sample included caregivers of wide child age range (2–10 years old), across which developmental and dietary differences exist [47]. However, child age was controlled for in all statistical models to account for potential developmental differences. Additionally, several factors in the models trended toward, but did not reach, statistical significance at the 0.05 level, which we hypothesize is related to the use of a small sample size. Furthermore, the use of diet screeners for caregiver and child diet, though common in this type of research, may lead to both underreporting and over reporting of intakes for certain food groups, resulting in an inaccurate representation of dietary patterns. However, the dietary assessment tools used in this study are validated and frequently used in the literature and national surveys.

Dietary behaviors are complex and are influenced by multiple factors. In this study we analyzed several important family- and household-level factors; however, it is important to note that there are other factors that may influence child dietary intake that were outside the scope of this study. For example, modeling behavior and dietary intakes of elder siblings, and family dietary restrictions or eating patterns (such as households following a vegan/vegetarian diet or avoiding certain foods due to food allergies) may also influence child dietary intake, and should be explored in future studies. Additionally, measurement and analysis of parenting styles as either direct predictors of child diet or potential moderators of the relationship between caregiver feeding practices and child diet were outside the scope of this study, but are important next steps in the research with this unique population. Child weight status is another factor that could be further explored. In this study, child height and weight were collected as caregiver-reported measures, and due to the limited accuracy of caregiver-reported height and weight observed in previous studies [48], BMI was not included in statistical models. Finally, Cronbach’s alpha values for some scales were low (modeling, child vegetable consumption, and less healthy home food availability scores) indicating potential for unreliability in the scale [49]. When indicated through statistical testing, steps were taken to increase Cronbach’s alpha values by removing items from scales. The scales used were drawn from validated measures that are commonly used in the literature, but because of low internal consistency of some scales, results should be interpreted with caution.

## 5. Conclusions

This study is among the first of its kind to assess caregiver feeding practices as predictors for child dietary intake among low-income, Appalachian communities and included an analysis of fruit and vegetable consumption as separate outcomes, which is a novel approach within this body of research. Caregiver modeling, healthy caregiver dietary intake, and healthful home food availability may promote healthier child dietary intake among young children in low-income, rural, Appalachian areas, which is consistent with previous findings in higher-income, urban, and more highly educated populations. Future research should consider differences in the relationship between caregiver feeding practices and fruit versus vegetable consumption, as certain feeding practices may influence the intake of these food groups differently. Further research is also needed to inform interventions to promote healthier child dietary intake by educating caregivers on the use and benefit of caregiver feeding practices that have a positive impact on child fruit, vegetable, and HS/HF snack food intake. 

## Figures and Tables

**Figure 1 nutrients-13-02773-f001:**
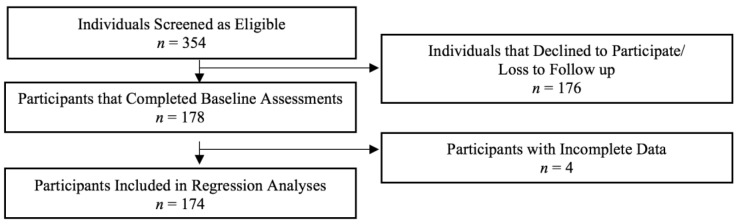
Sample Size and Participant Flow through Screening, Data Collection, and Analysis.

**Table 1 nutrients-13-02773-t001:** Characteristics of the Study Sample of rural, Appalachian Caregivers of young children (*n* = 178).

**Variable**	**Caregiver Age (** **±SD)**	**Child Age (** **±SD)**
Mean Age in Years (±SD)	35.6 ± 9.8	6.5 ± 2.7
	**Caregiver % (*n*)**	**Child % (*n*)**
Female	78 (138)	54 (96)
Race		
White	97 (173)	96 (171)
Not White ^a^	3 (5)	4 (7)
Ethnicity, % not H/L ^b^	99 (176)	98 (175)
Education		
Less than HS ^c^	16 (28)	
HS ^c^ or GED	53 (95)	
Some College	20 (35)	
College Degree or higher	11 (20)	
Marital Status		
Never Married	24 (43)	
Married	47 (83)	
Separated	8 (15)	
Divorced	17 (31)	
Widowed	3 (6)	
Income (in USD) ^d^		
$0–10,000	21 (38)	
$10,001–20,000	24 (43)	
$20,001–30,000	21 (37)	
$30,001–40,000	11 (19)	
$40,001–50,000	6 (11)	
$50,001–60,000	6 (11)	
$60,001+	7 (12)	
SNAP Participants ^e^	61 (109)	
WIC Participants ^f^	23 (41)	
Food Insecure Households	59 (103)	

^a^ Not White includes American Indian/Alaskan Native, Asian, Black/African American, and Other. ^b^ H/L refers to Hispanic or Latino. ^c^ HS refers to high school. ^d^ Four participants declined to provide income information. ^e^ SNAP refers to the Supplemental Nutrition Assistance Program. ^f^ WIC refers to the Special Supplemental Nutrition Program for Women, Infants, and Children.

**Table 2 nutrients-13-02773-t002:** Caregiver Modeling and Home Food Availability Scores.

**Modeling (*n* = 178)**	**Mean (SD)**
I try to eat healthy foods in front of my child, even if they are not my favorite ^a^	4.1 ± 0.94
My child learns to eat healthy snacks from me ^a^	3.9 ± 0.93
How often do you try not to eat unhealthy foods when your children are around? ^b^	3.2 ± 1.20
How often do you drink soda (regular or diet) or other sweetened beverages at meals and snacks with your child? ^b,c^	2.6 ± 1.30
I eat food I want my child to eat ^b^	4.2 ± 0.94
Modeling Scale Total Score	3.6 ± 0.63
**Home Food Availability Scores**	**Mean (SD)**
Healthier Home Food Availability ^d^ (*n* = 174)	20.3 ± 7.24
Less Healthy Home Food Availability ^e^ (*n* = 178)	10.1 ± 2.35

^a^ Responses were measured on a five-point Likert scale from strongly disagree—strongly agree. ^b^ Responses were measured on a five-point Likert scale from never—always. ^c^ Responses were reverse scored, per the scoring protocol of the validated survey instrument. ^d^ Healthier Home Food Availability is represented by a sum of the availability of 42 healthier foods in the household such as bananas, apples, carrots, low sugar cereals, whole grain bread, etc. ^e^ Less Healthy Home Food Availability is represented by a sum of the availability of 14 less healthy items in the household such as sugary cereals, cookies or candy, chips, soda, etc.

**Table 3 nutrients-13-02773-t003:** Self-Reported Caregiver and Child Dietary Intake.

**Child Dietary Intake (*n* = 178)**	**Mean Score (SD)**	**Possible Range of Scores**
Fruit ^a^	7.1 ± 1.78	0–8 (0 = never, 8 = 2+ times/day)
Vegetables ^b^	14.8 ± 4.70	0–24 (0 = never, 24 = 2+ times/day)
HS/HF snack foods ^c^	17.6 ± 6.33	0–32 (0 = never, 32 = 2+ times/day)
**Caregiver Dietary Intake (*n* = 178)**	**Mean Times per Day (SD)**	**--**
Fruit ^d^	1.0 ± 1.09	N/A
Vegetables ^e^	1.8 ± 1.16	N/A

^a^ The child fruit variable includes frequency of consumption of fresh, frozen, or canned fruit with possible scores ranging from 0–8, where 0 = never and 8 = 2 or more times per day. ^b^ The child vegetable variable includes a sum of the daily consumption of lettuce/green salads, non-fried potatoes, and “other” vegetables with possible scores ranging from 0–24, where 0 = never and 24 = a report of 2 or more times per day for each vegetable category. ^c^ The child HS/HF snack food variable includes a sum of the daily consumption of candy; doughnuts and pastries; cookies, cake, pie, and brownies; and ice cream or other frozen desserts with possible scores ranging from 0–32, where 0 = never and 32 = a report of 2 or more times per day for each of the HS/HF snack foods. ^d^ The caregiver fruit variable includes frequency of consumption of fresh, frozen, or canned fruit reported in the number of times per day. ^e^ The caregiver vegetable variable includes frequency of consumption of salads, non-fried potatoes, and “other” vegetables reported in the number of times per day.

**Table 4 nutrients-13-02773-t004:** Multiple Linear Regression Predicting Child Fruit Consumption ^a^.

Variable (*n* = 174)	Beta	*p*-Value	95% Confidence Interval
Modeling	0.27	0.17	−0.12 to 0.67
Caregiver fruit consumption	0.29	0.02 *	0.04 to 0.54
Healthier home food availability	0.06	0.002 *	0.02 to 0.09
Child age ^b^	−0.11	0.02 *	−0.21 to 0.02
Child gender ^c^	0.25	0.32	−0.25 to 0.76
Household income ^d^	−0.04	0.46	−0.15 to 0.07

* Indicates a statistically significant *p*-value. ^a^ Overall model significance *p* < 0.001 *; R square = 0.18. ^b^ Child age at time of survey was calculated using date of birth. ^c^ Child gender was coded as Male = 0 and Female = 1. ^d^ Household income was included in the model as a categorical variable in USD, categories were as follows: (0–10,000, 10,001–20,000, 20,001–30,000, 30,001–40,000, 40,001–50,000, 50,001–60,000, 60,001–70,000, 70,001–80,000, 80,001+).

**Table 5 nutrients-13-02773-t005:** Multiple Linear Regression Predicting Child Vegetable Consumption ^a^.

Variable (*n* = 174)	Beta	*p*-Value	95% Confidence Interval
Modeling	1.02	0.04 *	0.05 to 2.00
Caregiver vegetable consumption	1.56	<0.001 *	1.00 to 2.10
Healthier home food availability	0.09	0.05 *	0.001 to 0.18
Child age ^b^	0.03	0.79	−0.20 to 0.27
Child gender ^c^	1.50	0.02 *	0.24 to 2.75
Household income ^d^	−0.13	0.36	−0.39 to 0.14

* Indicates a statistically significant *p*-value. ^a^ Overall model significance *p* < 0.001 *; R square = 0.27. ^b^ Child age at time of survey was calculated using date of birth. ^c^ Child gender was coded as Male = 0 and Female = 1. ^d^ Household income was included in the model as a categorical variable in USD, categories were as follows: (0–10,000, 10,001–20,000, 20,001–30,000, 30,001–40,000, 40,001–50,000, 50,001–60,000, 60,001–70,000, 70,001–80,000, 80,001+).

**Table 6 nutrients-13-02773-t006:** Multiple Linear Regression Predicting Child HS/HF Snack Food Consumption ^a^.

Variable (*n* = 178)	Beta	*p*-Value	95% Confidence Interval
Modeling	−1.43	0.05 *	−2.85 to 0.00
Less healthy home food availability	0.59	0.003 *	0.20 to 0.98
Child age ^b^	−0.02	0.90	−0.37 to 0.32
Child gender ^c^	−1.12	0.23	−3.00 to 0.75
Household income ^d^	−0.18	0.35	−0.56 to 0.20

* Indicates a statistically significant *p*-value. ^a^ Overall model significance *p* < 0.001 *; R square = 0.10. ^b^ Child age at time of survey was calculated using date of birth and was reported as a continuous value. ^c^ Child gender was coded as Male = 0 and Female = 1. ^d^ Household income was included in the model as a categorical variable in USD, categories were as follows: (0–10,000, 10,001–20,000, 20,001–30,000, 30,001–40,000, 40,001–50,000, 50,001–60,000, 60,001–70,000, 70,001–80,000, 80,001+).

## Data Availability

The data presented in this study are available on request from the corresponding author. The data are not publicly available due to ongoing research being conducted with this dataset.

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
