# Peer review of "Caregiver Feeding Practices as Predictors for Child Dietary Intake in Low-Income, Appalachian Communities"

_nutrients, 2021, doi:10.3390/nu13082773_

Round 1
Reviewer 1 Report
The manuscript titled "Caregiver Feeding Practices as Predictors for Child Dietary Intake in Low-Income, Appalachian Communities" examines caregiver feeding (modeling and intake) practices and its relationship with child diet in low-income Appalachian families. While there are similar studies, this is the first of its kind looking at the specific Appalachian low-income community, and the first to look at modeling, dietary intake and home food availability all together. I found it particularly interesting that the authors conducted separate analyses on fruits and vegetables. Along with it being novel, this approach may help differentiate how intake of fruits and vegetables may have different contributing factors, not always related.
I just have a few questions about the study that I would like the authors' views on.
- What are the authors' views on the role of siblings, especially elder siblings as models? Was this part of the data captured? If so, do you think family size/number of household members would affect child dietary intake?
- Did the study have any families with dietary restrictions, like vegetarian/vegan or allergies to multiple common foods in the survey?
- It was not clear to me if the study design described is the same as SSTN, or an extension of SSTN. Is there a citation available, or are the authors describing what was done as part of SSTN itself?
- How do the authors differentiate between caregiver modeling and dietary intake? The difference to me is very subtle. Is this the difference between showing vs doing?
- What do the authors think about including child BMI in the models? Did you intentionally leave out child health parameters to focus on diet exclusively?
- Could the authors explain the fruit, vegetable and snack intake scoring of 0-8, 0-24, and 0-32 and its basis?
- What does modeling (higher score) mean in the context of unhealthy snack consumption - does it mean modeling (encouraging) eating less healthy/HS-HF foods?
- Can the authors comment on the percent of variability explained by the regression models? Is this typical of similar analyses? What according to you could be the major contributors confounding the effect and reducing the percent of variability explained, besides the low sample size?
- Did the authors see any correlations between poverty levels or food insecurity and availability of healthy/unhealthy food at home?
- line 328 - "..the general consensus being that among children, fruit consumption is easier to modify than vegetable consumption "- This could be due to the ready-to-eat nature of fruits, as compared to some cooking/processing required for vegetables. What are the authors’ thoughts?
- The Acknowledgement section is not edited from the default text, along with partial edits in the Funding section
Overall, the study is quite well laid out and analyzed, and extremely clear to follow. The results will lay foundations for future health planning, keeping the specifics of the community in mind.
Author Response
Please see the attached pdf for the response to Reviewer 1.

Reviewer 2 Report
Abstract
- I have some concerns with this section (1) the date of sampling took place and the age of participants should be clearly stated; (2) B and P-value should be included only; (3) please provide a clear conclusion with a broad summary of the study's implications and significance.
Introduction
- The authors provided review of studies as a rationale for the importance of this study topic. They reviewed studies on caregiver feeding practices influence on children's food choices but the age ranges were not clear.
- The term "caregiver" is unclear. Do you mean mother, father, grandparents…etc?
- Line 30-33: Please provide more detailed information about dietary recommendations for US children.
- Line 38-40: This systematic review contains little synthesis of the information. Please provide more information about the results of this review.
- Line 43-45: Very old reference- please update.
- Line 40, 60: Please clarify "populations" here. Do you mean lower/higher income populations?
- Line 45-56: Authors may want to answer the following questions: (1) How parents can influence children's food choices via their attitudes, beliefs, role-modelling, and style of parenting; (2) Is there any evidence to suggest that parent role modeling and styles are associated with children's food choices, and how? (Please refer to this article PLoS One. 2017, 24;12(5):e0178149). Parenting styles may act as a moderator of the association between parenting behaviors and children's dietary intake; (3) What is the difference between parenting style and parenting practice? Authors should expand on parenting style as a possible influencer on the changes in children's dietary intake. I would recommend referring to these articles (Ecol Food Nutr. 2015, 54, 93-113; Psychol Health. 2020, 35(11):1326-1345).
- Line 49-64: These paragraphs could benefit from more recent literature on a topic.
- Line 91-92, Line 95-96: "among families with young children in low-income, rural areas in Appalachian East Tennessee". Please use it once within a context.
Method
- I prefer to separate materials and methods into 3-4 separate sections. For examples, participants and study design, measures, data analysis…etc.
- Please refer to my comment in abstract. When caregivers were surveyed?
- The exclusion criteria should be clearly defined.
- Line 170: Data collection procedures should be described in sufficient detail. It is unclear to me how the data were collected from caregivers?
- Was a test for normality used before performing linear regression? The assumptions of linear regression were not stated (multicollinearity, homoscedasticity) (Line 186-187). How the covariates were enter in the model? Did you use a stepwise regression? Was it forward selection or backwards selection?
Results
- Line 192: A diagram is needed to show the final selection of participants.
- Is it necessary to include "declined to answer" in Table 1?
- 95% Confidence Interval for Beta should be included in Tables 4-6. I would suggest deleting standard error from all tables.
Discussion
- Description of studies is unclear (Lines 314,315,321,323,333,341,343,345,348). Please define these studies-cross-sectional or longitudinal.
- The conclusion section is missing. The scope for future research should be clearly mentioned.
Author Response
Please see the attached pdf for responses to Reviewer 2.

Round 2
Reviewer 2 Report
Dear Authors,
Well Done. The paper is significantly improved by these revisions. I have only one comment.
I would suggest moving P-value to the last column in Tables 4-6.